# Material Selection and Characterization for a Novel Frame-Integrated Curtain Wall

**DOI:** 10.3390/ma14081896

**Published:** 2021-04-10

**Authors:** Mercedes Gargallo, Belarmino Cordero, Alfonso Garcia-Santos

**Affiliations:** 1Department of Construction and Architectural Technology, Technical School of Architecture of Madrid, Technical University of Madrid (UPM), Av. Juan de Herrera, 4, 28040 Madrid, Spain; alfonso.garciasantos@upm.es; 2Arup Gulf Ltd., 39th Floor, Media One Tower, Dubai P.O. Box 212416, United Arab Emirates; 3Eumada FZ LLE, Creative Tower, Fujairah P.O. Box 4422, United Arab Emirates; belarmino.cordero@gmail.com

**Keywords:** composite glass structures, facades, high-rise buildings, curtain wall, GFRP, adhesive, glass fiber reinforced polymer, frame-integrated system

## Abstract

Curtain walls are the façade of choice in high-rise buildings and an indispensable element of architecture for a contemporary city. In conventional curtain walls, the glass panels are simply supported by the metal framing which transfers any imposed load to the building structure. The absence of composite action between glass and metal results in deep frames, protruding to the inside, occupying valuable space and causing visual disruption. In response to the limited performance of conventional systems, an innovative frame-integrated unitized curtain wall is proposed to reduce structural depth to one fifth (80%) allowing an inside flush finish and gaining nettable space. The novel curtain wall is achieved by bonding a pultruded glass fiber reinforced polymer (GFRP) frame to the glass producing a composite insulated glass unit (IGU). This paper selects the candidate frame and adhesive materials performing mechanical tests on GFRP pultrusions to characterize strength and elasticity and on GFRP-glass connections to identify failure module and strength. The material test results are used in a computer-based numerical model of a GFRP-glass composite unitized panel to predict the structural performance when subjected to realistic wind loads. The results confirm the reduction to one fifth is possible since the allowable deflections are within limits. It also indicates that the GFRP areas adjacent to the support might require reinforcing to reduce shear stresses.

## 1. Introduction

External envelopes are the image of every building creating fundamental component of the scenario of cities. The construction sector is constantly looking at the development of new building construction systems [1]. Traditional brick or heavy weight envelopes have been replaced for decades by lightweight enclosures, such as metal–glass facades and ventilated or rainscreen walls [2]. Metal and glass facades, known as curtain walls, are built from metal framing with the spaces filled with glass [3]. Nowadays curtain walls are an indispensable element of architecture for a contemporary city [4]. The two main curtain wall systems are stick-built and unitized [5]. The stick-built system was the initial curtain wall system with a metal framework of vertical mullions and horizontal transoms attached to the building and supporting glass panels installed on site [6]. Unitized curtain wall systems consist of cladding units where façade panels (typically glass, metal or stone) and metallic framing members (mullions and transoms) are pre-assembled in factory and then transported to site and attached to the load-bearing elements in the buildings, normally via pre-fixed brackets along the edge of the structural floor slab. Unitized curtain wall systems are the façade system of choice in high-rise buildings because the prefabricated assembly of units ensures high quality and allows fast installation without external access [7]. The current generation of unitized curtain wall systems is designed to transfer lateral loads, typically wind-induced pressures to the structural floor slabs [8]. This is achieved by the façade panels which are simply supported by the façade framing members which in turn transfer the loads by spanning between the floor slabs. 

### 1.1. Limitations of Conventional Curtain Walls 

Due to the difference in the thermal expansion coefficient of the glass and metal, conventional curtain wall systems require a flexible adhesive which limits any possibility of composite structural action between the glass and the framing.

The absence of composite action between the glass panels and the frames leads to deep profiles, that invade valuable space protrude to the inside and cause visual disruption. Moreover, the façade framing members are often made of aluminum alloys or other metals with characteristically high thermal conductivity, resulting in significant thermal transmission at façade framing members. This can be partly overcome by introducing thermal breaks in the façade framing members, but this increases the complexity and overall depth of the curtain wall.

### 1.2. State of the Art

Numerous investigations cover the mechanical properties of glass fiber reinforcement polymer GFRP [9,10]. Wurm [11] carried out a research that ended building three proof of concept mock-up of GFRP-glass composite units with two glass panels adhesively bonded to GFRP profiles located parallel along the glazing cavity. Petersen [12] pursued a window system with GFRP profiles bonded to an insulated glass unit (IGU) aiming for composite structural behavior and low thermal transmittance. Seele [13] investigated a frameless IGU with a cavity spacer bar structurally bonded to the glass unit. The GFRP-glass beam proposed by Bedon et al. [14,15] aimed to optimize the composite behavior of a beam consisted of two monolithic glass face sheets structurally bonded to GFRP pultruded core profile [16]. All of these previous research studies were focused on proof-of-concept prototypes. Limited investigations on the applicability on curtain wall systems or test results are available. 

The size-buckling of compressed Bernoulli–Euler nano-beams were investigated by stress-driven non-local continuum mechanism by Barretta et al. [17]. Furthermore, strategies to predict stiffening and dynamic responses of modern composite nano structure was proposed by Pinnola et al. [18]. Non-linear analyses of laminated functionally graded-graphene platelet-reinforced beams resting on an elastic foundation and based on two-phase stress-driven non-local model were investigated by Ansari et al. [19]. However, there are a no sufficient validated analytical or mathematical models of the mechanical response of GFRP-glass composite units to extract conclusions.

In the construction area a large range of adhesives and sealants are used, such as polyurethane, epoxies, polyamides, ethylene-vinyl acetate-copolymers, acrylates, poly(vinyl acetate)s, silicones, etc. [20]. Adhesives can also be found at a wide variety of strength levels depending on the necessity of their applications [21]. Five candidate adhesives were studied for load bearing steel-glass connections. On this study, Overend [22] carried out mechanical testing and mathematical modelling to assess the feasibility of the adhesive connections. Comparable protocol was undertaken by Nhamoinesu and Overend [23] who studied adhesives for a steel-glass composite unit that could be applied on façade system. Previous research covering adhesive for glass bonding that helped as a basis for this study were: (a) Belis et al. [24], who studied silicones, polyurethanes, MS-polymers, acrylates and epoxies to bond glass to metal, (b) Peters [25] that focuses his research on adhesive for glass and fiberglass and (c) Blues et al. [26] who researched the bond of glass and metal focused on load transmission and failure behavior for façade system applications.

### 1.3. Proposed Design

A conventional curtain wall system with deep framing protruding from the glazing unit is represented in Figure 1. The proposed GFRP-glass composite unitized system is based on a GFRP E-profile adhesively bonded along the four edge of an IGU as sketched in Figure 2. The absence of the metal framing decreases the thermal transmittance and the use of a stiff adhesive activates the composite behavior between the glass and the GFRP frame. Moreover, the GFRP profile is placed within the IGU cavity width avoiding any frame projecting out from the glazing unit. 

Preliminary calculations were carried out to assess the structural feasibility of the initial design. These calculations indicated that the proposed system could achieve the same wind pressure defection criteria as the conventional non-composite system, but it could do so in one fifth of the depth required by the conventional system. However, the shear stresses generated in the GFRP pultrusion were almost three-times higher than the design shear stresses recommended by the manufacturers. These preliminary calculations were instrumental to identify the system variables such as framing depth, width, adhesive thickness and GFRP web thickness. However, the preliminary structural analysis was based on Euler–Bernoulli simple bending theory, i.e., shear deformations across the depth of the composite unit and shear lag across the width of the composite panel are ignored. Moreover, the preliminary calculations assume that the materials are linear elastic thereby ignoring the time and temperature dependent properties of the adhesives and the GFRP. It is therefore pertinent to characterize the nonlinear response of the materials and to use these properties in a non-linear finite element analysis of a unitized composite GFRP–glass panel.

The novelty of the composite unitized curtain wall system studied in this research paper is that the framing is pultruded (GFRP) with a coefficient of thermal expansion of similar value to the glass. This similarity allows the use of stiffer adhesives with thinner bond lines that activates the composite action between glass panels and frames. The supplementary benefit of the pultruded GFRP is a lower thermal conductivity compared to aluminum reducing heat transfer and the risk of condensation [27].

### 1.4. Study Aims

In response to the limited performance of conventional systems, an innovative frame-integrated unitized curtain wall is proposed to reduce structural depth significantly, allow an inside flush finish and reduce thermal transmission at joints. The proposed design integrates the principles of composite structural action into a slim unitized curtain wall aiming a more efficient use of materials to reduce structural depth. 

This paper investigates and characterizes candidate frame and adhesive materials for this novel frame-integrated unitized curtain wall through mechanical testing and subsequently used this material-level test data in a numerical model of a GFRP-glass composite unitized panel subjected to realistic loads.

## 2. Materials and Methods

The novel curtain wall is achieved by adhesively bonding a pultruded GFRP frame to the edge of flat glass panels thereby producing a composite insulated glass unit (IGU). Figure 3 indicates the methodology followed for testing and result assessment. Four-point bending tests are performed on candidate GFRP pultrusions to characterize shear strength and modulus of elasticity. These are then followed by single-lap shear tests on GFRP-glass connections to select the adhesive.by identifying failure module and strength. The results provided by the mechanical tests are fed into a numerical model to predict the structural performance of the proposed system.

### 2.1. Materials

#### 2.1.1. GFRP Specimens

For the selection of GFRP specimens, the variables investigated were (i) the composition of matrix; polyester or phenolic resins [28] and (ii) the effect of elevated temperatures [29]. A total of 40 pultruded GFRP bars manufactured in accordance with BS EN 13706-1:2002 [30] and measuring 150 mm × 20 mm× 5 mm were tested as per Table 1. The glass fibers were aligned in the longitudinal direction in all the specimens. 

Heat soaking was applied to some of the specimen to simulate any variation that could occur by being exposed to solar radiation or high temperatures during the curing. Those specimens were placed in a kiln at 130 °C for 30 min and were left to cool down at ambient conditions prior to testing. 

#### 2.1.2. Adhesive Specimens

The selection of the candidate adhesive for this study was based on the available manufacturer’s technical data Fiberline [31]; Huntsman [32]; 3M Scotch-Weld [33]; Dow Corning [34], and previous research studies on bonding adhesive [22,24,25,26,35]. In addition, several considerations were taken during the selection of the material. Candidate adhesives should:Have a shear strength in the range between 5 and 10 MPa based on preliminary wind load analytical calculations.Be minimum 2 mm, since the minimum permissible thickness of the bond was based on the fabrication allowable tolerances of GFRP and glass [36].Have a reasonable durability when exposed to UV radiation and moisture [37,38].Be dimensionally stable against moisture changes.Maintain not less than 75% of their shear strength at a temperature of +80 °C. The reason is that at elevated temperatures some adhesives might lose stiffness and strength [39].

Based on the above a range of acrylate, epoxies and silicone were selected for this study. Different thicknesses were also considered as listed in Table 2.

#### 2.1.3. Glass for Testing

The glass used in the single lap shear test was toughened glass panels in accordance with to BS EN 12150-2:2004 [40]. Glass panel size was 300 by 300 mm and the thickness was 10 mm. Glass panels were the same in all specimens.

### 2.2. Methods

#### 2.2.1. Four-Point Bending Test

The equipment for the four-bending test was based on ASTM D7264/D7264M—07 [41]. The testing apparatus was an Instron 5567 (Instron, Norwood, MA, USA) with a 30 kN load cell at a loading rate of 2 mm per minute. Two round supports with same height were used to place the GFRP bars. The supports were located at a 135 mm distance and with the center aligned with the center line of the crosshead connected to the testing apparatus. The crossheads were located at a 75 mm distance. The center of the GFRP bars were clamped with a steel plate. Two displacement gauges were used to measure the deflection of the GFRP bars: first gauge measuring the steel plate displacement and second at the crosshead. The setup of the four-point bending test is shown in Figure 4.

The gauges recorded displacements every 0.25 s and the modules of elasticity was calculated for each measurement. The shear strength was calculated as per Equation (1):(1)$$\tau_{\mathrm{beam}}=\frac{{\mathrm{VAcy}}^{\prime }}{\mathrm{Ia}}$$
where τ_beam_ is the shear stress of the beam at a certain point, V is the shear force, Ac is the section area over the cut line, y′ is the span from center of area over the cut line to the centroid of the total section, I is the second moment of area of total section and a is the width of the section at the cut line.

The modulus of elasticity was calculated as per Equation (2):(2)$$E=\frac{\mathrm{MR}}{I}$$
where E is the modulus of elasticity, M is the applied moment, I is the second moment of area and R is the radius of curvature. 

#### 2.2.2. Single Lap Shear Test

The equipment used for the single lap shear test was in accordance with ASTM D1002 [42]. The apparatus was an Instron 5500R (Instron, Norwood, MA, USA) with a 150 kN load cell. A glass panel was used with two GFRP bars adhered to two opposite sides of the panel. The bars were clamped to the apparatus and to displacement gauges. An L-shape steel plate was bonded at 80 mm from the glass edge with the gauge probe touching the plate. The gauges measured the vertical displacement for each adhesive joint. To avoid measuring the elongation of the GFRP bars the displacement gauge were fixed at the inner edge closed to the steel place. The test set up is shown in Figure 5.

The adhesive application protocol was the same, including curing temperature and pressure for all the candidate adhesive except for with the exception of the TSSA, which followed manufacturer’s recommendations [34]. The candidate adhesive was applied on a similar. Shims were used to administer the correct thickness for each application.

The testing equipment induced a 0.2 mm in-plane displacement per minute until failure. At failure the load, extension and shear stress were recorded. The displacement was recorded at 0.2 mm per minute until failure. 

#### 2.2.3. Computer Verification Method

##### Software and Model

The glass–GFRP composite unit was modelled in a finite element analysis (FEA) software named LUSAS v14.5 (LUSAS, Kingston upon Thames, UK). LUSAS is a software developed for the analysis of composite products and components. It allows for static and modal dynamic analysis using beam, shell, solid and joint elements, and composite elements. Figure 6a shows the geometry of the non-linear elastic model assessed. A four-node tetrahedral element type which is a 3-dimensional isoparametric finite element with linear interpolation order was used.

The glass-GFRP composite unit was considered symmetrical at both axis (x and y). Therefore, only a quarter of the unit was modelled. An out-of-plane restraint was added at point B of the GFRP E-profile indicated in Figure 6a assuming that would be the connection bracket to the primary structure. Symmetrical boundary conditions were applied on the yz-plane at the CD edge and on the xz-plane at AD edge (Figure 6a).

The glass panels were modelled using a mesh with a 10 mm thickness and divided in four elements. The GFRP E-profile was modelled using a 5 mm thickness mesh and divided in one element, same as the adhesive but with a 2 mm thickness (Figure 6b). The GFPR and adhesive were modelled in smaller divisions due to the expected larger displacements and higher stress gradients. 

The output is considered at each element node and Gauss point. Each node and point had both direct and shear stresses and strains values. Results are calculated based on the constitutive relationship at the element Gauss points. Extrapolation is carried out to calculate the nodal stresses from the Gauss points as per Equations (3) and (4): (3)$${\underline{\sigma }}_{i}\;\left(\xi_{i},\eta_{i}\right)=\overset{N}{\underset{I=1}{\sum }}N_{I}\left(\xi_{i},\eta_{i}\right){\underline{\sigma }}_{I}$$
(4)$${\underline{\varepsilon }}_{i}\;\left(\xi_{i},\eta_{i}\right)=\overset{N}{\underset{I=1}{\sum }}N_{I}\left(\xi_{i},\eta_{i}\right){\underline{\varepsilon }}_{I}$$
where *N* is the number of Gauss points, *i* is the nodal point values, *I* is the Gauss point value.

##### Applied Load

Wind load is generally the dominant load for curtain walling which may vary depending on load duration. For this study, two wind load duration were chosen: a high load for a short period (Load case 1) and a low load for a long period (Load case 2). The aim was to assess any effect on the alteration of the modulus of elasticity of the GFRP and adhesive based on load duration. 

Building codes are frequently used to deliver wind load pressures on façades. The aim of the codes is to interpret the dynamic action of wind and converted into a static action for load calculation. The building codes relay on basic wind speed with several factors applied specific to each building: gust effects, internal pressures, building height, etc. The basic wind speed varies depending on the code and location. 

BS EN 1991-1-4 [43] assumes a 10 min mean wind velocity with an annual risk of being exceeded of 0.02. The 10 min mean wind velocity is considered the main basic pressure, while the peak velocity pressure is based on a one second load duration pressure. The formulas given in the National Annex (NA) 2.17 [43] are used in this study to calculate the one second gust (5) and for 10 min wind (6) in a town terrain:q_p_ = C_e_(z)·C_e_,_T_·q_b_(5)
q_p_ = q_b_(6)
where q_p_ is peak velocity pressure, q_b_ is mean basic velocity pressure. C_e_(z) is value of exposure factor in accordance with NA.7 [43], C_e,T_ is value of exposure correction in accordance with NA.8 [43].

C_e_(z) and C_e,T_ depend on the distance from coast line and the building height. As per code [43] the basic velocity pressure is obtained at 100 m above ground. For this study 100 m height and a location 10 km from the coast line have been assumed to estimate the wind pressure. The proportion between basic velocity pressure and peak velocity pressure has been assumed as 4 as follows:10-min wind (q_p_ = 750 N/m^2^) is Load case 11-s gust (q_p_ = 3000 N/m^2^) is Load case 2

Load cases 1 and 2 were uniformly applied in the FE as *q* on the top surface of the glass panel (Figure 6a). 

##### Material Properties for Computer Verification

The three materials modelled in the FEA analysis were the glass, the GFRP and the adhesive. While a linear elastic material was assigned to the glass, the GFRP and adhesive were attributed as elastic-perfectly plastic materials. The software used could address non-linearity geometric shape. Therefore, it is considered that GFRP-Glass composite unit was assessed as a non-linear-elastic model. 

Table 3 summarizes the material mechanical properties inputted in the FEA analysis for load case 1 and 2 as described in Section Applied Load.

## 3. Results

### 3.1. Four-Point Bending Test

During the four-point bending test all specimens delaminated due to horizontal shear stress as indicated in Figure 7a for polyester resin GFRP bars and in Figure 7b for the phenolic resin bars. 

The specimen taken obtained an analogous shear strength mainly between 17 MPa to 19 MPa as shown in Figure 8. This average shear strength was below the 25 MPa initially provided by the manufacturer [31].

The modulus of elasticity calculated is summarized in Figure 9. Long duration loads provided a modulus of elasticity similar to the values given by the manufacturer [31] in the rage of 23 MPa and 30 MPa. However, for the short loads, the modules of elasticity obtained was twice and sometime three times the value of the long duration loading as can be seen in in Figure 9. Based on this result, it is consequential to note that the modulus of elasticity of GFRP varies considerable based on load duration.

Based on the results obtained, it was concluded that there was not significant variance on the mechanical properties of the variables studied: polyester or phenolic resins and the effect of elevated temperatures.

### 3.2. Single Lab Shear Test

The failure of mode and mean shear strength of the adhesive were recorded during the testing regime and are summarized in Table 4 and short commentary describing the findings on each adhesive follows.

3M Scotch Weld DP 490 with a thickness of 3 mm: Glass failure was observed in all specimens. Mainly glass breakage and one specimen due to paring of the glass at the edge. It was considered that adhesive peak shear stress was caused by differential shear [44]. These stresses could have been transferred to the glass leading to high local stress concentration at the edge as the basis of the failure.3M Scotch Weld DP 490 with a thickness of 5 mm: The thicker specimens of the DP490 ended also with glass failure. Although in this case mainly due to glass plucking. These adhesive specimens achieve the highest mean shear strength.3M Scotch Weld 2216 B/A with a thickness of 3 mm: The failure mode and shear strength values shown by this adhesive during the single lab shear test was inconsistent with former studies [23].Dow Corning TSSA with a thickness of 3 mm: The tested shear strength of the adhesive was 0.26 MPa which is considerably lower than expected according to the information provided by the manufacturer [34] and was rejected.Huntsman Araldite 2047 with a thickness of 3 mm: Adhesion failure was observed in all specimens, leading to conclude that the GFRP bar surface required a level of roughness to be introduced.Huntsman Araldite 2047 with a thickness of 3 mm thick and abraded GFRP: Significant enhancement of the results were obtained when compared to the non-abraded GFRP and same adhesive. It is to be noted that this adhesive presented a plastic deformation before failure.

Figure 10 illustrates results obtained during the single lab shear test for each of the candidate adhesive, except for the Dow Corning TSSA that was rejected. The 3M Scotch-Weld DP 490 bond with a thickness of 5 mm obtained highest mean shear strength. It is to be noted the increase of flexibility by enlarging the thickness from 3 mm to 5 mm. The abraded version of the Huntsman Araldite 2047 bond also provided plausible load bearing capacity and showed a certain plastic deformation prior to failure. The 3M Scotch Weld 2216 B/A did not provide sufficient shear strength and was discarded along with the Dow Corning TSSA. 

### 3.3. Computer Verification of the Proposed System

#### 3.3.1. Glass Deflection 

Based on the CWCT Standard for systemized building envelopes [45] clause 3.5.2.5, the allowable deflection limit was 15mm at the middle of the IGU edge. 

The deflection produced by both Load cases 1 and 2 as illustrated in Figure 11.

Glazing deflection results for (a) Load case 1 and (b) Load case 2 Figure 11 were within the limit. Utilization ratio for the deflection were of 18% and 51% for Load cases 1 and 2 respectively.

#### 3.3.2. Glass Tensile Stress 

ASTM E 1300 [46] provides the limit of surface stress for heat strengthened glass and toughness glass as 46.6 MPa and 93.1 MPa, respectively. 

From the FEA analysis results shown in Figure 12, it can be observed that the highest tensile stress was located at the mid-pan of the glass long edge. The tensile stress produced by both Load cases 1 and 2 as illustrated in Figure 12 were within the limit. The utilization ratio was 9% and 31% for Load cases 1 and 2 respectively for the heat strengthened glass and consequently lower for toughness glass.

#### 3.3.3. Adhesive Shear Stress 

The adhesive shear stress limit was set by the testing results obtained in Section 3.2. The mean shear stress at failure for the Huntsman Araldite 2047 with abraded GFRP was of 3.57 MPa and for the 3M Scotch Weld DP 490 with a 5 mm thickness was 4.70 MPa. 

The results shown in Figure 13 indicates that both Load cases 1 and 2 were within the limits set by single lad shear test results and with a utilization ratio of 16% and 55% respectively for the Huntsman Araldite 2047 and consequently lower for the 3M Scotch Weld DP 490.

#### 3.3.4. GFRP Shear Stress

The GFRP shear stress limit was set by the four-point bending testing results obtained in Section 3.1, which was considered 17 MPa. The FEA model results indicated that the GFRP E-profile might fail due to shear when subject to short duration load case with shear stress 2.5-times larger than those obtained in the testing. The location of the failure of the GFRP profile was close to the supporting point as indicated in Figure 14.

It is worth mentioning that the shear strength obtained in the FEA analysis in the longitudinal direction was much lower than at corners. This lower shear strength is due because the fibers are set mainly along the longitudinal axis. Allowing the fibers to be in various directions might have the benefit of increasing the shear strength of the bars.

## 4. Discussion

### 4.1. Selection of GFRP Material for Framing

The results of the four-bending test shown that there were not considerable differences in the mechanical properties of the variables studied. Therefore, it was decided that polyester matrix would be used for the GFRP framing as it has a finer aspect and it is more affordable. It was also concluded that the heat soaking was unnecessary. 

The modules of elasticity obtained for the short duration of loading was twice and three-times higher than the value for the long duration loading. This significant variation dictates that it is fundamental to determinate the duration of the loading of which the GFRP frame would be subjected to. The wind loading considered in this study is in accordance with BS EN 1991-1-4 [43] which assumes the short duration loads have a factor of 4 with respect to the long duration load cases.

All specimens obtained an analogous shear strength which was in all cases below the value provided by the manufacturer [31]. The FEA analysis indicated that the GFRP bar might fail close to the supporting point when subjected to the short duration loads. This indicates that the vulnerability of the GFRP-glass composite unit is the shear strength at the corner support and increasing the strength at that area requires to be studied further. Possible enhancement could be adding steel plates at the corners or allowing the glass fibers to be located in various directions during the pultrusion [47].

### 4.2. Selection of Adhesive Material for Bonding

Following assessment was made for each candidate adhesive during the selection:3M Scotch-Weld DP 490 with a thickness of 3 mm: Based on the majority of glass breakage failures, it was concluded that this adhesive was a very stiff. To increase the flexibility of the bond a 5 mm thick adhesive was used in subsequent test.3M Scotch-Weld DP 490 with a thickness of 5 mm: Glass plucking was the main failure more observed with the increased thickness of this adhesive. It was concluded that a load path eccentricity might have occurred and led to bending moment at the glass edge. Since the mean shear strength was the highest, it makes this adhesive a potential candidate for the GFRP-glass composite unit.3M Scotch Weld 2216 B/A with a thickness of 3 mm: The failure mode and shear strength values shown by this adhesive during the single lab shear test was inconsistent with former studies [23]. In addition, the mean shear strength obtained was low and therefore this adhesive was discarded for the GFRP-glass composite unit.Dow Corning TSSA with a thickness of 3 mm: The low shear strength obtained when compared to value provided by the manufacturer led to conclude the application of this adhesive was not carried out in accordance with manufacturer’s recommendations [34]. This adhesive was discarded for the GFRP-glass composite unit until further testing in line with manufacturer’s recommendations are executed.Huntsman Araldite 2047 with a thickness of 3 mm: Since adhesion failure was observed in all specimens, it was decided to carry out the test with abraded GFRP. The results of the non-abraded GFRP were discarded.Huntsman Araldite 2047 with a thickness of 3 mm thick and abraded GFRP: Significant enhancement were observed when compared to non-abraded GFRP results for the same adhesive. Before failure, the adhesive presented a plastic deformation that could be beneficial in the GFRP-glass composite unit as it might give a visual deformation as a signal of a future collapse.

Based on the above, it was concluded that both 3M Scotch Weld DP 490 and Huntsman Araldite 2047 are potential adhesive for the GFRP-glass composite unit.

### 4.3. Further Investigations

Based on the results obtained a series of future investigations could be contemplated:GFRP framing: Investigating options to reinforce the GFRP at the areas adjacent to the supports. A solution could be reinforcing with steel plates or by arranging the glass fibers in multiple directions in the pultrusion fabrication process [47].Load duration: Carry out testing at full scale prototype subject to dynamic wind pressure to investigate the variable modulus of elasticity obtained in the four-point bending test.Adhesive: Further testing of 3M Scotch-Weld DP 490 and Huntsman Araldite 2047 could be carried out to explore the suitable thickness for each adhesive. While a 5 mm thick bond is appropriate for DP 490, the thickness of Huntsman Araldite 2047 could be reduced to 2 mm to increase its stiffness. TSSA could be re-tested ensuring that the adhesive storage and application protocol follow manufacturer’s recommendation. For all candidate adhesives, it would be good to eliminate or reduce the induced bending moment by clamping the GFRP close to the bond or performing double-lap instead of single-lap shear tests.

## 5. Conclusions

This research concludes that the reduction of curtain wall structural depth to almost one fifth compared to conventional curtain wall systems is possible. The concluded statement is based on the following results obtained from the mechanical tests and computer verification:The maximum deflection at the glass edge, maximum tensile stress at the glass surface and maximum shear stress at the adhesive connection are within admissible values and below 55% utilization ratio.The shear stress at the GFRP frame was generally within allowable values except at areas adjacent to the support, which are currently 2.5-times larger than the limit set by the four-point bending test. This indicates that the vulnerability of the GFRP-glass composite unit is the shear strength at the corner support and that increasing the strength at that area requires to be studied further.The short duration load is decisive for the feasibility of the GFRP-glass composite unit. The wind loading considered in this study was in accordance with BS EN 1991-1-4 [43] which assumes the short duration loads have a factor of 4 with respect to the long duration load cases. For future research, it is advisable to test a full scale prototype subject to dynamic wind pressure to investigate the modulus of elasticity.

## Figures and Tables

**Figure 1 materials-14-01896-f001:**
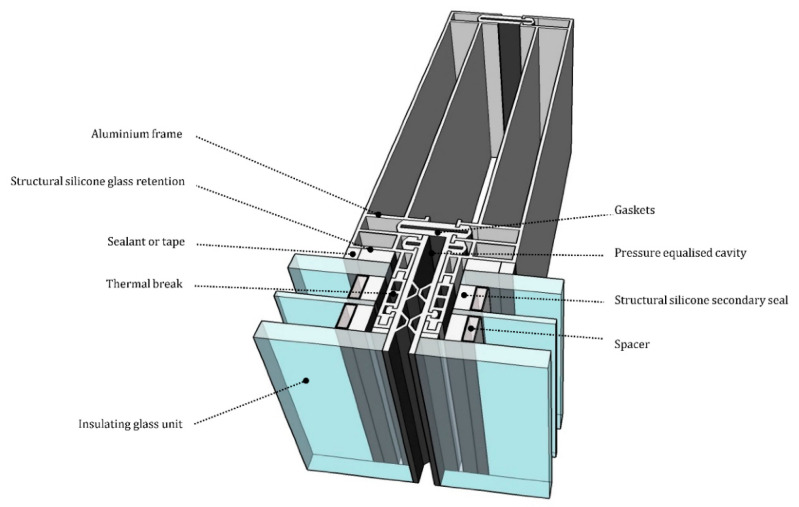
Section through mullion of conventional unitized curtain wall system with triple glazed insulated unit.

**Figure 2 materials-14-01896-f002:**
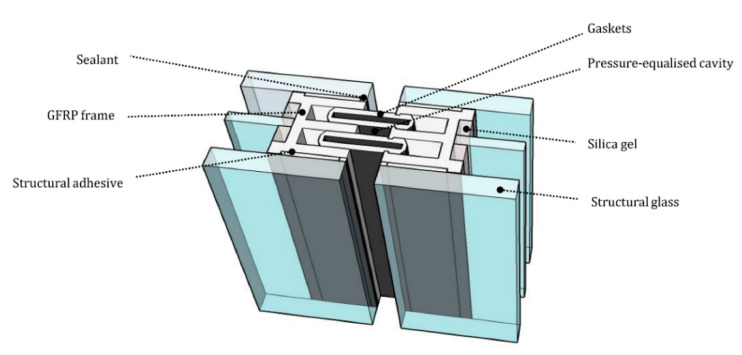
Section through mullion of the proposed system with triple glazed insulated unit.

**Figure 3 materials-14-01896-f003:**
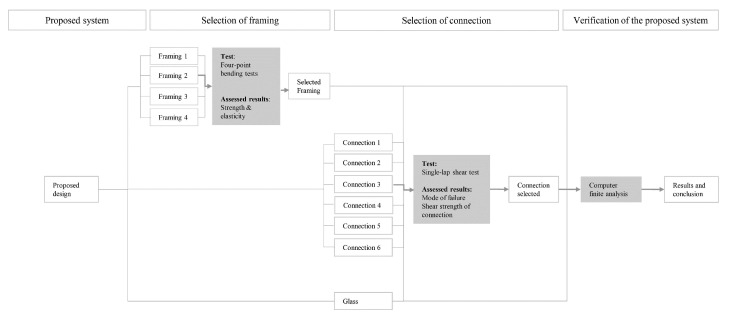
Methodology for testing and result assessment.

**Figure 4 materials-14-01896-f004:**
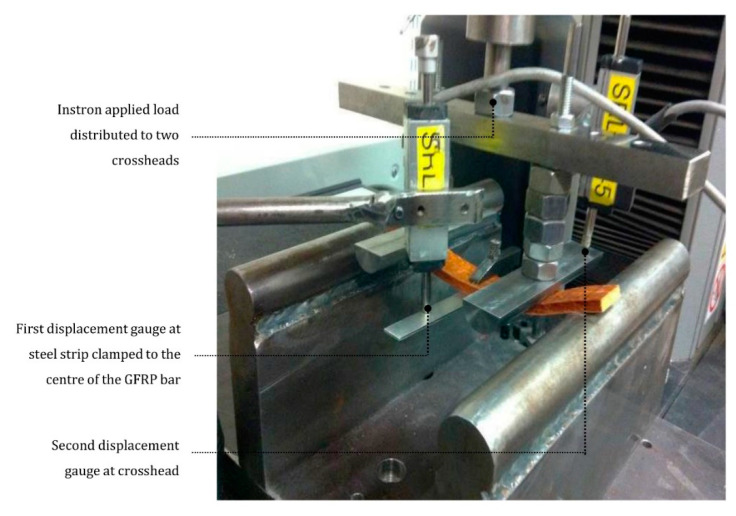
Specimen being tested on Instron 5567 machine.

**Figure 5 materials-14-01896-f005:**
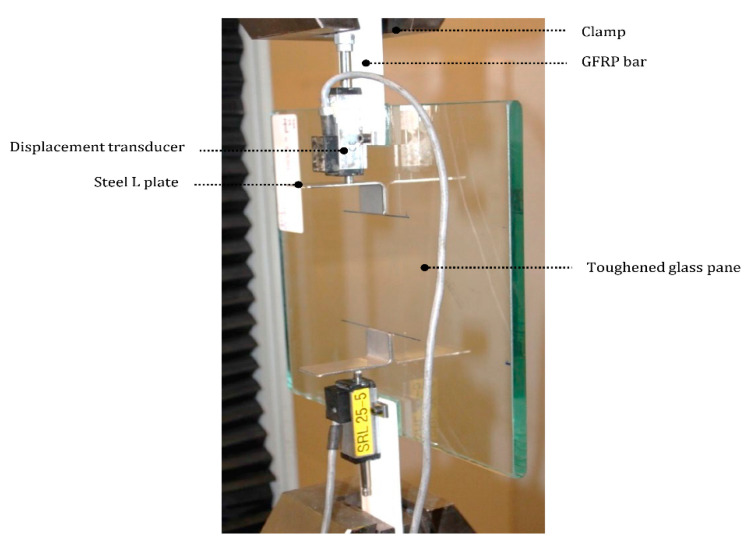
Specimen being tested on Instron 5500R machine.

**Figure 6 materials-14-01896-f006:**
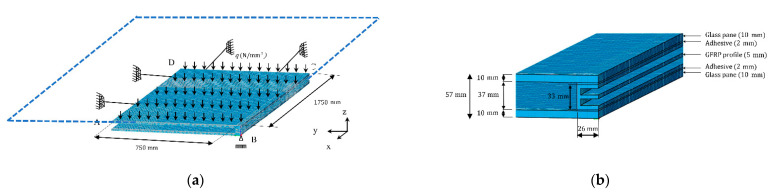
Glass–glass fiber reinforced polymer (GFRP) composite unit entire model (**a**) and (**b**) closed view of the joint.

**Figure 7 materials-14-01896-f007:**
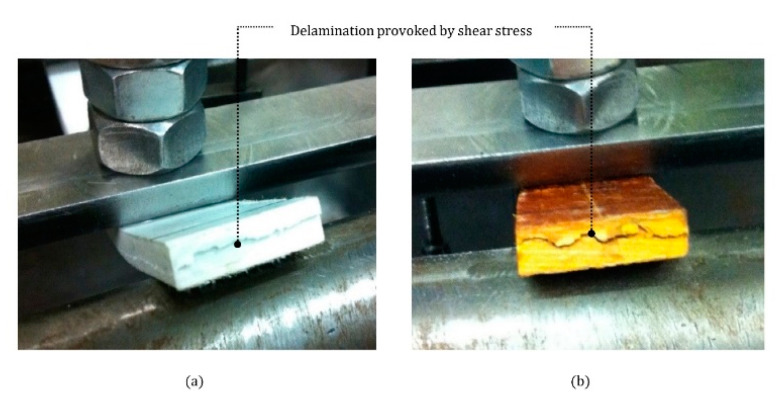
Horizontal shear stress failure in (**a**) polyester resin and (**b**) phenolic resin specimens.

**Figure 8 materials-14-01896-f008:**
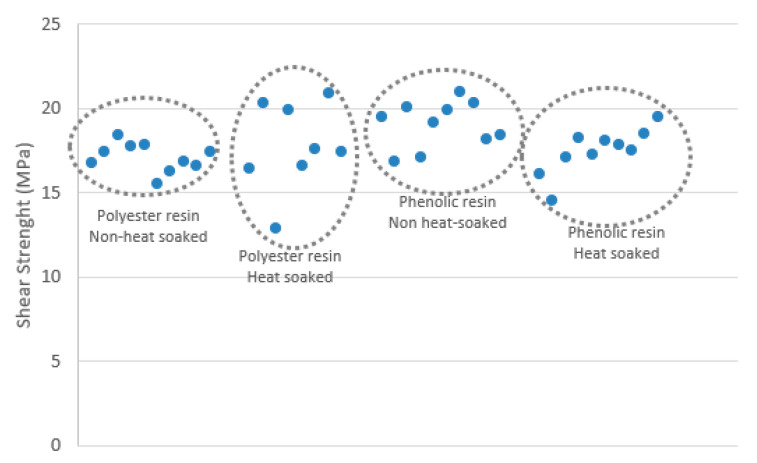
Shear strength obtained for each variable of GFRP specimens.

**Figure 9 materials-14-01896-f009:**
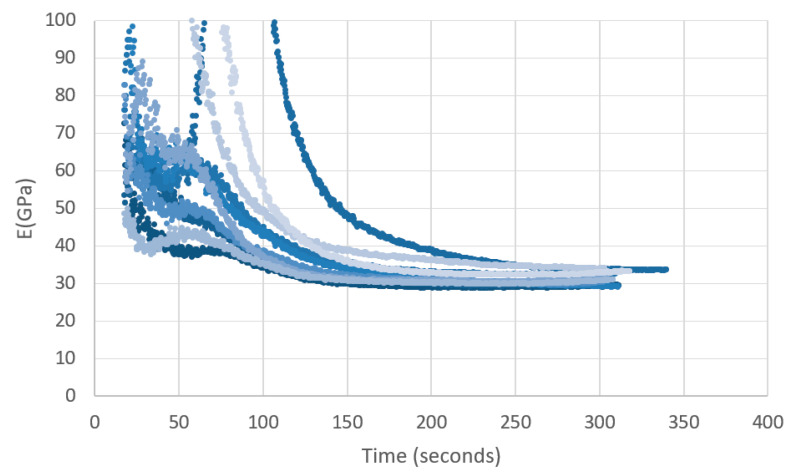
Modulus of elasticity obtained for each specimen based on duration of loading.

**Figure 10 materials-14-01896-f010:**
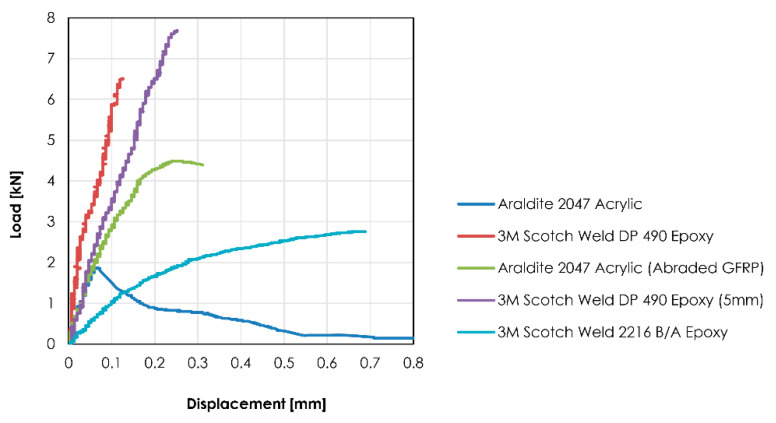
Load vs. displacement curves of candidate adhesive.

**Figure 11 materials-14-01896-f011:**
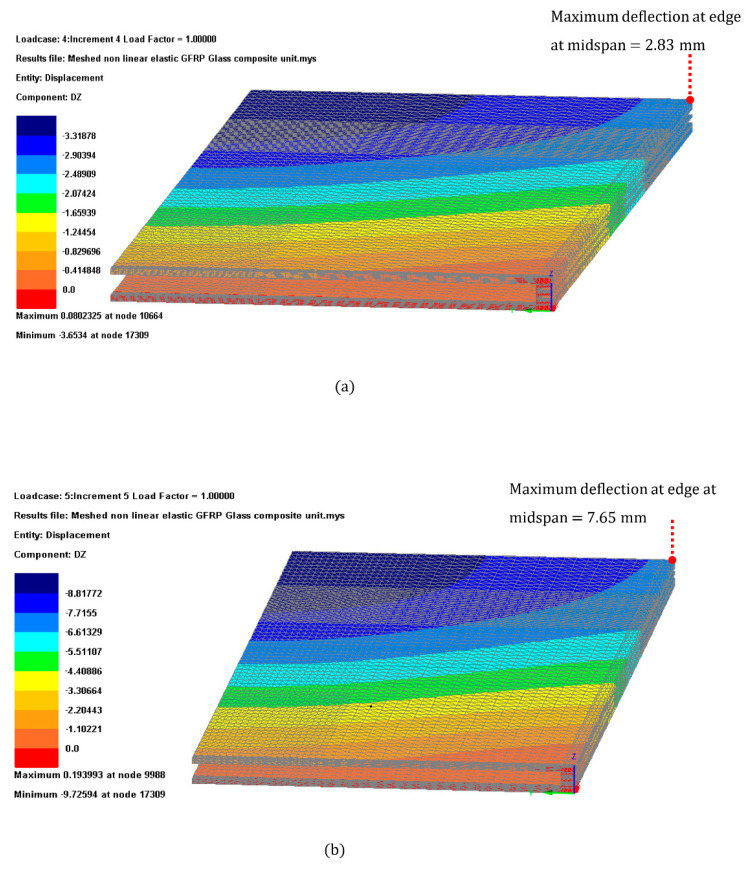
Glazing deflection results for (**a**) Load case 1 and (**b**) Load case 2.

**Figure 12 materials-14-01896-f012:**
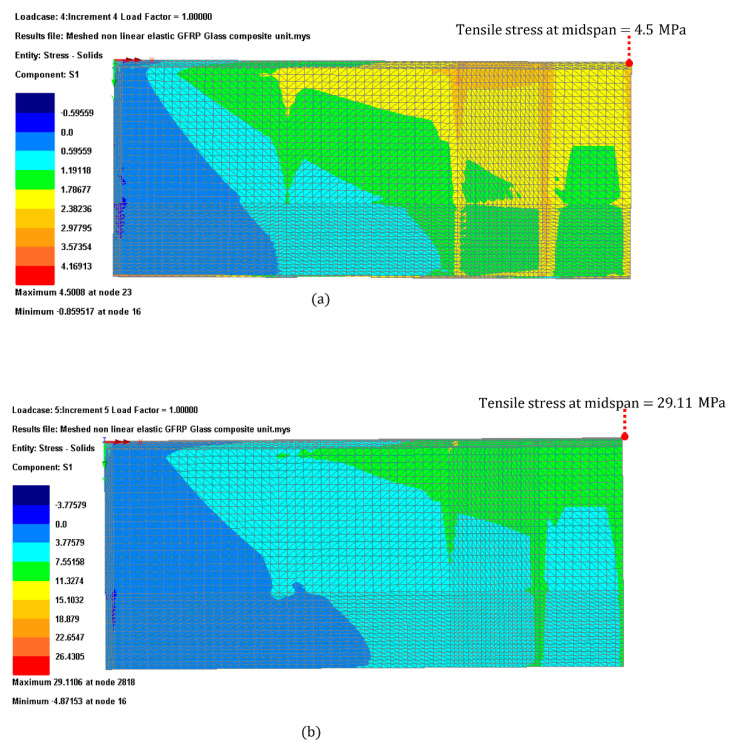
Tension stress contour plot for (**a**) Load case 1 and (**b**) Load case 2.

**Figure 13 materials-14-01896-f013:**
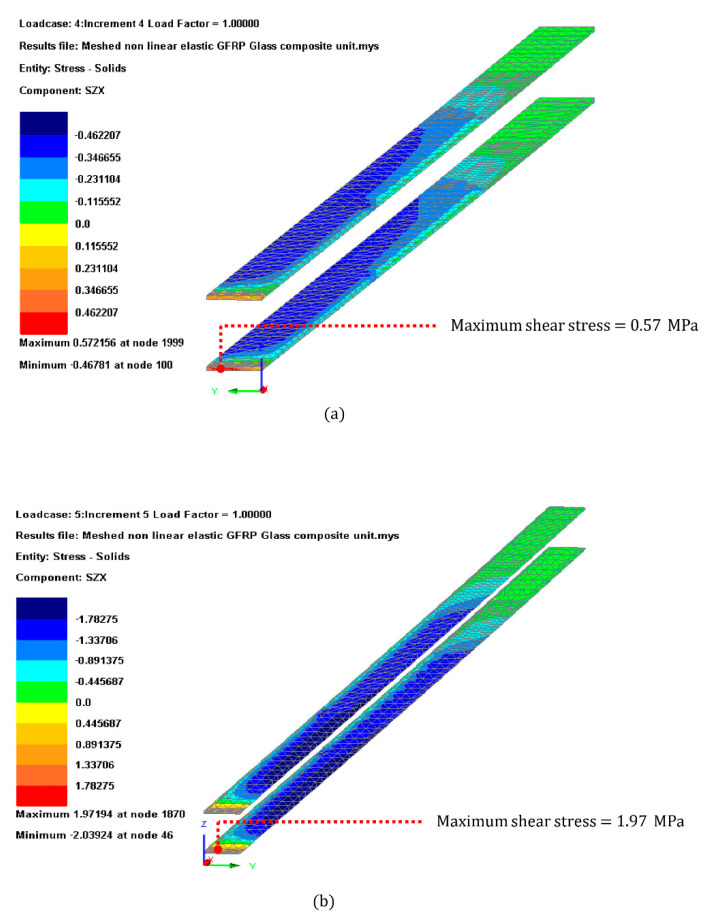
Adhesive shear stress contour plot for (**a**) Load case 1 (**b**) Load case 2.

**Figure 14 materials-14-01896-f014:**
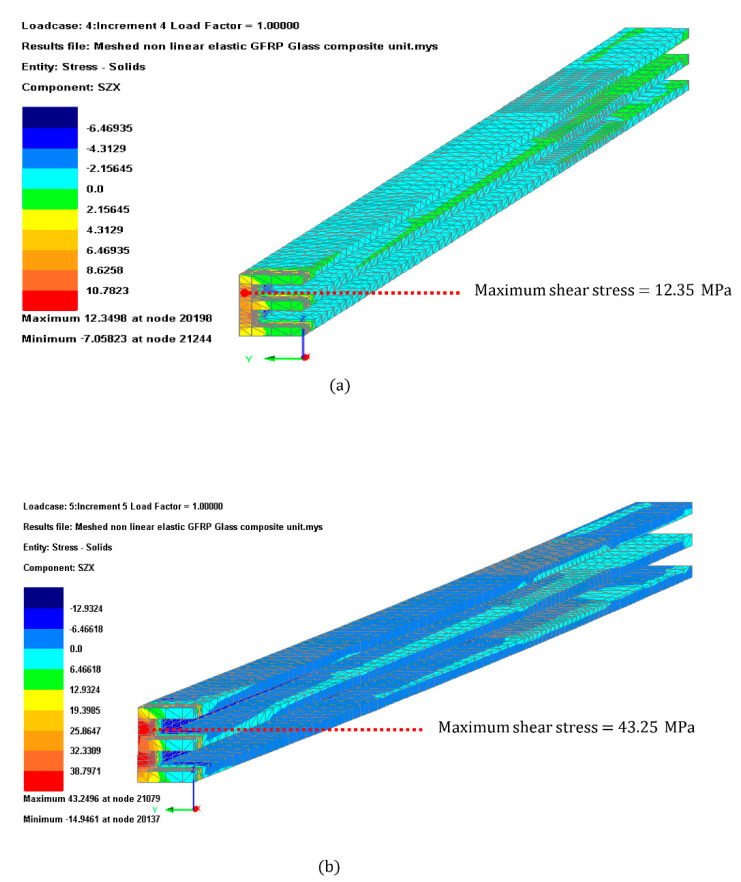
GFRP shear stress contour plot for (**a**) Load case 1 (**b**) Load case 2.

**Table 1 materials-14-01896-t001:** Number of specimens for each variable set.

Matrix	Polyester Resin	Phenolic Resin
Heat soaked	10	10
Non-heat soaked	10	10

**Table 2 materials-14-01896-t002:** Selected candidate adhesive products for single lab shear test.

Adhesive Type	Brand	Product Name	Thickness
Acrylate	Huntsman	Araldite 2047	3 mm
Epoxy	3M Scotch	Weld DP 490	3 mm
Epoxy	3M Scotch	Weld DP 490	5 mm
Epoxy	3M Scotch	Weld 2216 B/A	3 mm
Silicone	Dow Corning	Transparent Silicone Structural Adhesive (TSSA)	3 mm

**Table 3 materials-14-01896-t003:** Mechanical properties of materials.

Mechanical Property	Load Case 1	Load Case 2
Wind load duration	600 s	1 s
Wind pressure	750 N/m^2^	3000 N/m^2^
Glass E	70 GPa	70 GPa
Glass v	0.23	0.23
GFRP E	23.37 GPa ^1^	100 GPa ^2^
GFRP v	0.3	0.3
GFRP yield stress	75 MPa	75 MPa
Huntsman Araldite A2047 E	142.35 GPa ^3^	634.90 GPa ^4^
Huntsman Araldite A2047 v	0.43 ^4^	0.43 ^4^
Hunstman Araldite yield stress	3.61 ^5^	3.61 ^5^

^1^ Obtained from four point bending test (Section 3.1). ^2^ Value assumed from t = 1 as initial loading instead of t = 0. ^3^ Approximated by extrapolation. ^4^ Extracted from Nhamoinesu research of adhesive for steel-glass connection [23]. ^5^ Extracted from BSI reinforced plastics composites [43].

**Table 4 materials-14-01896-t004:** Summary of results single lap shear tests.

Candidate Adhesive	Thickness	Total Number of Specimens	Failure Mode	Number of Specimens with Each Failure Mode	Mean Load at Failure	Mean Displacement at Failure	Mean Shear Strength
3M Scotch Weld DP 490	3 mm	10	Breakage of glass Paring of glass at joint	9 1	5.38 kN	0.11 mm	4.49 MPa
3M Scotch Weld DP 490	5 mm	3	Breakage of glass Plucking of glass	1 2	5.64 kN	0.18 mm	4.70 MPa
3M Scotch Weld 2216 B/A	3 mm	3	Adhesive failure Plucking of glass	2 1	2.26 kN	0.46 mm	1.88 MPa
Dow Corning TSSA	3 mm	10	Cohesion failure Adhesive failure	9 1	0.26 kN	24.09 mm	0.21 MPa
Huntsman Araldite 2047	3 mm	10	Adhesion failure	10	1.32 kN	0.81 mm	1.10 MPa
Huntsman Araldite 2047 with Abraded GFRP	3 mm	3	Paring of glass at joint Adhesion failure	2 1	4.28 kN	0.28 mm	3.57 MPa

## Data Availability

The data presented in this study are available on request from the corresponding author.

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
