# Peer review of "Material Selection and Characterization for a Novel Frame-Integrated Curtain Wall"

_materials, 2021, doi:10.3390/ma14081896_

Round 1

Reviewer 1 Report

In this manuscript authors show characterization of materials for a frame-integrated curtain wall. The information shown in the manuscript are interesting. The characterization methods are well chosen.

My recommendation is: accept

Regards       

Author Response

Dear reviewer,

Thank you very much for taking the time to review our paper.

The manuscript has been updated to reduce the similarity index and incorporate other peer reviewer’s comments. 

We hope the revised manuscript still found acceptable from your side.

Best regards,

Mercedes Gargallo 

Reviewer 2 Report

The manuscript “Material selection and characterization for a novel frame-integrated curtain wall” is related to an actual problem of selecting reliable materials for civil engineering, in particular for conventional curtain walls.

Nowadays, the most spread design is based on the glass panels supported by the metal framing which transfer imposed loads to the building structure. The proposed curtain design is realized through bonding a pultruded Glass Fiber Reinforced Polymer (GFRP) frame to the glass producing a composite Insulated Glass Unit. The study is oriented towards selecting candidate frame and adhesive materials through mechanical tests on GFRP pultrusions. In doing so, strength and elasticity and on GFRP-glass connections are estimated with identification of failure module and strength.

The state of the art is clearly characterized.

The design of the proposed curtain wall is duly described.

Materials and methods are characterized in due details.

The manuscript contains a lot of interesting and important results including both experimental evidences and computational data. The manuscript is technically correct and is of due length.

The manuscript is summarized with a conclusion. It requires some revision.

In general, the manuscript contains new and important results. However, it looks more like a technical report rather than research paper.

The manuscript is very inaccurately prepared.

Pages 11, 12. The table 3 is given twice.

Page 13. Figure 8. The data on elastic modulus should be given with the scatter information.

Page 13. Figure 9. Right figure. The Power fit does not look appropriate and should be changed.

Page 13. Figure 9. There are too many curves on both graphs. This means large scatter of the properties. It is not clear, how this information might be correctly interpreted.

Page 14. The page is mostly empty, while the sentence is terminated without an end.

Page 20. The second sentence is terminated without an end. The reason for using bold font is not clear.

Page 23. It is postulated that: “Based on the results obtained from the mechanical test and computer verification it can be concluded that the reduction of structural depth to almost one fifth compared to benchmark conventional system is possible”. The phrase does not look being correctly formulated.

Page 24. The conclusion. Since the manuscript deals with the design aspects, the numerical values should be given and achieved results are to be deeply generalized.

Author Response

Dear reviewer,

Thank you very much for taking the time to review our manuscript and for providing valuable comments.

The paper has been revised to incorporate your comments. A point by point response has been provided in the attached file for easy reference. The manuscript has also been updated to reduce the similarity index.

We hope the revised paper is found acceptable from your side.

Best regards,

Mercedes Gargallo

Reviewer 3 Report

The topic is interesting and the manuscript is organized with figures and diagrams.
Some comments are listed as follows:

1.    Several components of the proposed element have a small thickness. Hence, in order to enrich the Introduction in the framework of mechanics of small-scale structures, some recent relevant papers on nonlocal models can be considered such as
Buckling: Buckling loads of nano-beams in stress-driven nonlocal elasticity (2020) Mechanics of Advanced Materials and Structures, 27 (11), pp. 869-875.
Gradient nonlocality: Variationally consistent dynamics of nonlocal gradient elastic beams (2020) International Journal of Engineering Science, 149, art. no. 103220;
New stress driven model: Nonlinear analysis of laminated FG-GPLRC beams resting on an elastic foundation based on the two-phase stress-driven nonlocal model. Acta Mech (2021). https://doi.org/10.1007/s00707-021-02935-4 and so on.
2.    A FE analysis is performed in Section 3.3. Some more information should be added on the implemented analysis in order to point out if a commercial code has been used and if the small-scale of some components have been taken into account.

Author Response

Dear reviewer,

Thank you very much for taking the time to review our manuscript and for providing valuable comments.

The paper has been revised to incorporate your comments. A point by point response has been provided in the attached file for easy reference. The manuscript has also been updated to reduce the similarity index.

We hope the revised manuscript is found acceptable from your side.

Best regards,

Mercedes Gargallo 

Round 2

Reviewer 3 Report

References must be carefully checked (authors' name, volume of the Journal, pages, article namber, etc.) since there are slips, see e.g.:

ref [15] C. P. A. A. L.-N. M. O. F. F. Chiara Bedon ...

ref [17] C. M. J. O. M. Pascual ...

ref [18] F. F. R. L. F. M. d. S. G. R. R. Barretta ...

ref [19] S. A. F. R. B. F. M. d. S. Francesco P. Pinnola ...

ref [20] M. F. O. M. R. H. R. R. Ansari ...

ref [21] L. A. B. M. A. C. F. A. R. J. F. C. M. N. Solange Magalhaes ...

ref [27] C. S. M. A. T. U. H. F. T. V. Martin Blues ...

Author Response

Dear reviewer,

Thank you very much for taking the time to review our revised manuscript. References have been updated to include all authors’ names, journal, volumes, etc) as follows:

Ref: 15. Numerical investigation on structural glass beams with GFRP-embedded rods, including effects of pre-stress. Bedon, C., Louter, C. 2018, Composite Structures, Vol. 184, pp. 650-661.

Ref: 17. Buckling loads of nano-beams in stress-driven nonlocal elasticity. R. Barretta, F. Fabbrocino, R. Luciano, F. Marotti de Sciarra, G. Ruta. 11, 2020, Mechanics of Advance Materials and Structures, Vol. 27, pp. 869-875. DOI: 10.1080/15376494.2018.1501523.

Ref: 18. Variationally consistent dynamics of nonlocal gradient elastic beams. Francesco P. Pinnola, S. Ali Faghidian, Raffaele Barretta, Francesco Marotti de Sciarra. 2020, International Journal of Engineering Science Applied Physics, Vol. 149. DOI: 10.1016/j.ijengsci.2020.103220.

Ref: 19. Nonlinear analysis of laminated FG-GPLRC beams resting on an elastic foundation based on the two-phase stress-driven nonlocal model. R. Ansari, M. Faraji Oskouie, M. Roghani, H. Rouhi. s.l. : Springer Professional, 2021, Acta Mechanica, Vol. 2.102. DOI: 10.1007/s00707-021-02935-4.

Ref 20. Brief Overview on Bio-Based Adhesives and Sealants. Solange Magalhaes, Luis Alves, Bruno Medronho, Ana C Fonseca, Anabela Romano, Jorge F.J. Coelho, Magnus Norgren. 2019, Polymers.

Ref 21. Mechanical Behavior of Toughened Epoxy Structural Adhesives for Impact Applications. Gamze S Bas, Erol Sancaktar. 2020, ChemEngineering.

Ref 27. Thermal performance of novel frame-integrated unitised curtain wall. Belarmino Cordero, Alfonso Garcia-Santos, Mauro Overend. 2015, Revista de la Construccion, Vol. 14, pp. 23-31.

We hope the article is found acceptable from your side.

Best regards,

Mercedes Gargallo